# Influence of Modified PVA Fiber on Ultra-High Performance Concrete and Its Enhancing Mechanism

**DOI:** 10.3390/polym16233449

**Published:** 2024-12-09

**Authors:** Zhiyuan Chen, Hongyu Fan, Wanying Zheng, Siheng Zhang, Xi Wu, Tengfei Fu, Demei Yu

**Affiliations:** 1College of Transportation and Civil Engineering, Fujian Agriculture and Forestry University, Fuzhou 350108, China; 2Department of Civil Engineering, Hangzhou City University, Hangzhou 310015, China; 3Zhejiang Engineering Research Center of Intelligent Urban Infrastructure, Hangzhou City University, Hangzhou 310015, China

**Keywords:** polyvinyl alcohol fiber, UHPC, enhancing mechanism

## Abstract

In this study, the properties of ultra-high-performance concrete (UHPC) were enhanced by adding modified polyvinyl alcohol (PVA) fibers. The specimens with different curing ages were evaluated in various aspects to investigate the effects of different dosages, lengths, and surface treatments of PVA fibers on the performance of UHPC. The performance was compared with that of steel fiber-reinforced UHPC with the same ratio and multiple dosages. At the same time, the distribution of fibers and the morphology of fibers were observed by a scanning electron microscope, and the mechanism of fiber reinforcement was discussed. The results showed that the mechanical properties were significantly affected by the fiber dosage, length, and surface treatment. Based on the test results, the optimum PVA fiber addition can increase the compressive strength and flexural strength by 12.0% and 6.0% compared to the control UHPC without fibers. A comprehensive evaluation was carried out and indicated that the optimum PVA fiber addition has the potential to replace 0.5% steel fiber in certain conditions.

## 1. Introduction

Ultra-high-performance concrete (UHPC), which is a novel construction material with superior mechanical properties as well as high durability, has shown great advantages in the field applications of highway and marine engineering [1,2,3]. Steel fiber is a key component to improve the mechanical properties, ductility, and crack resistance of UHPC [4,5,6,7,8,9]. However, Choi et al. found that although the addition of steel fibers improved the mechanical strength of UHPC, the specific strength (ratio of compressive strength to density) did not always increase due to increased density [10]. Wang et al. and Zhang et al. showed that steel fibers at the surface were susceptible to corrosion, which might affect the long-term performance of UHPC [11,12]. In addition, steel fiber has a high carbon emission and cost, which results in limitations in UHPC applications [13,14,15]. Therefore, many scholars have carried out research on replacing steel fibers with eco-friendly synthetic fibers.

As an alternative synthetic fiber, polyvinyl alcohol (PVA) fiber has been extensively used in fiber-reinforced concrete [16,17,18,19,20]. Zhang et al. [16] showed that due to the different strengths of interfacial connection between fibers and the matrix, the strength of concrete increased with the dosage of steel fibers but decreased with the dosage of polypropylene fibers (PPFs). Xu et al. [17] showed that the interfacial bonding of PVA fibers affects stress transfer in concrete, and effective interfacial treatment of the fibers can improve the macro-mechanical properties. Shiferaw et al. [18] studied the effect of different PVA fiber sizes on the bridging effect, and the results showed that during the pull-out test, fibers with smaller diameter fractures and fibers with larger diameters were pulled out. PVA fiber with a 27 μm diameter was optimum to enhance the mechanical properties. Du et al. [19] reported that with the increase of PVA fiber dosage and crack width in fiber-reinforced cementitious composites was significantly reduced with prolonged crack initiation and formation of a through crack. The presence of PVA fiber reduced the energy release rate during cracking. Shabakhty et al. [20] found that while glass fiber mainly increased strength, PP fiber and PVA enabled the engineered cementitious composites with high ductility. 

The shortcomings of UHPC reinforced by steel fiber, such as brittleness, tensile strength, and insufficient crack control ability, have hindered its further development [21]. Moreover, the embedded steel fibers may have a high risk of corrosion under extreme weather conditions [22]. Lin et al. [23] replaced 33.3% of steel fibers with paraformaldehyde fibers (POMFs) to improve the compressive strength and toughness of UHPC. Li et al. [24] investigated the synergy of various synthetic fibers with steel fibers on the properties of UHPC, and the results showed that the toughening and cracking-resistant effects were achieved when the steel fibers and PVA fibers were mixed with 1.5% and 1.0% of steel fibers, respectively. 

To further increase the compatibility between PVA fiber and the cementitious matrix, the effect of the surface treatment was investigated [25,26,27,28]. Han et al. [25] showed that PVA fiber could increase the hydration degree of aerogel-incorporated concrete due to Ca^2+^ adsorption. Li et al. [26] reported enhanced mechanical strength, toughness, and pore structure of foam concrete reinforced by polar-agent-modified PVA fibers. Yao et al. [27] applied graphene oxide to the surface of PVA fiber and enabled better chemical bonds for hydration products. An enhanced interface was responsible for the improved toughness of cementitious composites. An in situ mineralization method was used by Han et al. [28] to form mineral layers of aragonite and calcite on the surface of PVA fiber. The failure of the interface changed from adhesion to cohesion, which was the reason for increased fiber pull-out force, mechanical properties, and energy absorption. In addition, Liu et al. [29] applied PVA fibers in 3D-printed concrete and found effective enhancement of the fracture performance.

Due to the low strength of PVA fiber, it is difficult to completely replace steel fiber in most cases. However, the low density and high ductility might enable PVA fibers to outperform steel fiber-reinforced UHPC in certain field applications, including easier mixing and casting, higher specific strength and better crack control, and 3D-printed UHPC. In this study, the performance of different dosages, lengths, and surface treatments of PVA fibers was investigated through a comprehensive evaluation, comparing the two kinds of fibers and exploring the possibility of PVA fiber as an alternative to steel fiber.

## 2. Materials and Methods

### 2.1. Materials and Mixture Proportions

A local P·I 52.5 ordinary Portland cement (3.15 g/m^3^) was used along with silica fume (2.37 g/m^3^) and quartz sand (2.67 g/m^3^) to prepare UHPC. The chemical compositions of cement, silica fume, and quartz sand are shown in Table 1. Particle size distributions are shown in Figure 1. The nominal maximum grain size for quartz sand is 0.6 mm. 

A steel fiber (7.85 g/m^3^) with a 13 mm length and a 220 μm diameter was used in the control UHPC mixture. The PVA fibers (1.3 g/m^3^), from a series of commercially available synthetic fiber products designed to be used in concrete, were provided by a local manufacturer (Baohualin Industry, Yongan, China). In total, four different types of PVA fibers were investigated in this study, namely, Type I, Type II, Type III, and Type IV. Type I PVA fiber was the unmodified product, whereas the other three types were modified using proprietary non-ionic surface treatment agents by the manufacturer to increase the density of the hydroxyl groups on the surface. As a result, Type II PVA fiber had the highest hydroxyl group density. The physical properties and dimensions of the fibers used are shown in Table 2. Images of the materials used are shown in Figure 2.

The mixture proportion of control UHPC is shown in Table 3. The mixture proportion of control UHPC was designed according to dense packing theory (a detailed description can be found in the literature [30]), with a fixed cement-to-silica fume volume ratio of 3:1 and a water-to-binder ratio of 0.13. A polycarboxylate superplasticizer was used to adjust workability, which was measured using a “mini” bronze slump cone (60 mm in height with a 36 mm top-opening diameter and a 60 mm bottom diameter). Superplasticizers were added as necessary to reach at least a 150 mm spread size for all mixtures.

For steel fibers, four different dosages were used, including 0.5%, 1.0%, 1.5%, and 2.0%. For PVA fibers, in total, nine mixtures were cast using different fiber types, lengths, and dosages (0.5%, 1.0%, and 1.5%). Table 4 shows the detailed fiber dosages. All fibers were added volumetrically. 

### 2.2. Experimental

#### 2.2.1. Specimen Preparation and Curing

UHPC specimens were prepared using a 5 L mortar mixture. First, all dry materials, including fibers, were added and mixed for 4 mins. Then, water and superplasticizer were added to the mixing bowl and mixed for another 8 mins. After the freshly mixed UHPC reached the desired workability (minimum 150 mm spread size in the mini-slump test), it was placed in 40 mm × 40 mm × 160 mm steel molds pre-coated with mold release oil and consolidated on a vibration table for one minute. The specimens were sealed and cured for 24 h and then transferred to a curing chamber (23 ± 2 °C, 95% RH) until the desired age for testing. 

#### 2.2.2. Mechanical Strength Testing

At ages of 3 d, 7 d, and 28 d, compressive strength and flexural strength were tested using 40 mm cubes and 40 mm × 40 mm × 160 mm prisms, respectively. Flexural and compressive strength tests were conducted at loading rates of 50 N/s ± 10 N/s and 2400 N/s ± 200 N/s, respectively. The maximum load at failure was recorded, and the force-displacement data during compression were also collected.

#### 2.2.3. BSE-SEM Microstructure Analysis

Specimens containing PVA fibers and steel fibers from the II 12 mm 0.5 and SF 0.5 groups were tested at 28 d. These specimens were cut into 10 mm × 10 mm × 5 mm pieces and then placed in molds. Resin and a curing agent were prepared at a mass ratio of 2:1 and poured into the molds under a vacuum environment to fully immerse the samples. After the resin was cured, the specimens were demolded and polished using sandpapers of varying grits from coarse to fine until a smooth and flat surface was obtained. The samples were then washed with anhydrous ethanol, dried, and used for backscatter SEM microstructure analysis.

#### 2.2.4. SE-SEM Microstructure Analysis

Specimens from the II 12 mm 0.5 and SF 0.5 groups at 28 d were subjected to flexural testing. The fracture surfaces of the broken specimens were sputtered and then used for secondary electron SEM microstructure analysis. 

## 3. Results

### 3.1. Effect of PVA Fiber on Mechanical Properties

#### 3.1.1. PVA Fiber Dosage

PVA fiber dosage determines the number of fibers in a given volume of UHPC. Too low of a fiber dosage cannot effectively control the cracks, limiting energy absorption at failure. On the other hand, too high of a fiber dosage will lead to a decline in mechanical properties due to poor consolidation, stress concentration, and insufficient bonding between the fiber and the mortar. An optimum dosage of fiber should exist so that the fibers can effectively distribute the stress, inhibit the propagation of cracks, and improve the mechanical strength. 

The 6 mm Type I PVA fibers were added to the specimens at 0.5%, 1.0%, and 1.5%. Figure 3 shows the compressive strength and flexural strengths at 3 d, 7 d, and 28 d. The results showed that at an early age (3 d and 7 d), the mechanical strength was adversely affected by the addition of fibers, which was likely due to hydration retardation caused by increased superplasticizer dosage to disperse the fibers. However, the compressive and flexural strength at 28 d were recovered to similar values of the control mixture (0% fiber dosage). In addition, Figure 3 shows that fiber dosages of more than 0.5% could cause a slight decrease in both compressive strength and flexural strength.

#### 3.1.2. PVA Fiber Length

The length of the PVA fibers could affect the crack-bridging effect as well as the energy absorption. Shorter fibers cannot effectively bridge cracks with limited reinforcement. Therefore, the 6 mm PVA fibers demonstrated the lowest mechanical properties. On the other hand, longer fibers may lead to intertwined fiber interactions and hinder the reinforcing performance. Optimum fiber length can make the fibers uniformly dispersed in the UHPC matrix and provide an effective bridging effect and energy absorption, thus improving mechanical strength. 

Type I fibers of a 0.5% dosage with different lengths of 6 mm, 8 mm, 12 mm, and 24 mm were added to UHPC specimens. The compressive strength and flexural strength at the corresponding ages are shown in Figure 4. Compared to Figure 3, it can be seen that at a 0.5% fiber dosage, all four different lengths of fibers enhanced the compressive strength and the flexural strength. In addition, a 12 mm fiber length demonstrates the highest increase of 5.7% and 10.6% in compressive strength and flexural strength, respectively. Therefore, 12 mm was determined to be the optimal length of fibers in this study.

#### 3.1.3. PVA Fiber Treatment

Effective surface treatment of PVA fibers can improve the interfacial bonding between fibers and the UHPC matrix, which will assist stress transfer and distribution. Insufficient interfacial bonding could cause the fiber to be pulled out prematurely under stress, limiting the crack-bridging effect and energy absorption. The results shown in Figure 5 demonstrate the effect of different fiber treatments. It can be seen that the addition of four types of fibers increased the mechanical strength of UHPC at a length of 12 mm with a fixed 0.5% fiber dosage. Compared with the specimens with unmodified fiber (Type I), the mechanical strength was further improved by Type II fibers. The flexural strength was almost the same as that of Type II fibers, and the compressive strength was increased by 5.9%. The compressive strength of Type III fibers was slightly increased by 1.8%, and the flexural strength was decreased by 4.9%. The compressive and flexural strengths of the specimens with Type IV fibers were both decreased. It was then determined that the 12 mm Type II fiber at a 0.5% dosage achieved optimum performance among all mixtures, with 158.97 MPa in compressive strength and 24.86 MPa in flexural strength at 28 d, which were 12.0% and 10.0% higher than the control UHPC without fiber. The optimum enhancement is likely due to the higher density of the hydroxyl group in Type II fiber because of the surface treatment. This result is consistent with Yao et al. [27], in which an optimum compressive strength enhancement was from a mixture with 0.5% PVA fibers of 12 mm in length. The enhancement was attributed to the high bonding strength between PVA fibers and the cementitious matrix. 

### 3.2. PVA Fiber Versus Steel Fiber

Steel fibers were added at 0.5%, 1.0%, 1.5%, and 2.0%, and the mechanical strengths are shown in Figure 6. Compared to the control UHPC, the addition of steel fibers can significantly improve the compressive strength. The compressive strength of steel fiber at a 2.0% dosage reached 186.83 MPa with a 17.5% increase. However, it should be noted that at a 0.5% dosage, PVA fiber (12 mm Type II) demonstrated similar performance to steel fiber. This indicates that at lower dosages, steel fiber does not have a significant advantage over modified PVA fiber. 

Figure 7 shows the specific strengths (ratio of compressive strength to density) of all mixtures. The specific strengths were improved in all steel fiber mixtures and most PVA fiber mixtures. It is also worth noting that, although the mechanical properties of Type II PVA fiber and steel fiber-reinforced UHPC are similar at lower dosages (0.5%). This result indicates that it is possible to use PVA fiber in lieu of steel fiber at lower fiber dosages.

### 3.3. Energy Absorption

Based on the force and displacement data collected during the compressive test, the stress–strain curves of all ratios were calculated, and the energy of destruction by the specimen in reaching the peak load was obtained by integrating the area underneath the stress–strain curves (Figure 8). Then, the toughness of the concrete can be quantitatively compared to the energy of destruction [31]. In Figure 8, it can be seen that the stresses in all fiber-reinforced concretes are greater compared to the control group at the same strain, which indicates that the addition of fibers improves the load-bearing capacity of the concrete. Meanwhile, it can be seen in Figure 9 that the energy absorption capacity of fiber-reinforced concrete is also improved, and the energy absorption capacity of steel fibers for concrete is much higher than that of PVA fibers. In the PVA fiber-reinforced concrete (0.5%, I 8 mm, I 12 mm, II 12 mm, and III 12 mm), the energy of destruction has been effectively improved compared with the control by 17.4%, 18.2%, 17.8%, and 18.6%, respectively. The energy of destruction of the steel fiber-reinforced concrete (0.5%, 1.0%, 1.5%, and 2.0%) was increased by 36.4%, 62.8%, 65.2%, and 105.9%, respectively. Overall, steel fibers improve the toughness of the concrete much more effectively than PVA fibers, which is due to the high strength of steel fibers that can make the concrete withstand greater deformation and load before failure. Nevertheless, the difference between the PVA fiber and steel fiber is narrowed down at a lower dosage.

### 3.4. Microstructure Analysis

Two groups of II 12 mm 0.5 and SF 2.0 specimen sections were selected, and the BSE-SEM images and EDS mapping of the specimens are shown in Figure 10. 

Based on the EDS mapping results combined with the diameter of the fibers, it can be distinguished that the steel fiber is the circular section in Figure 10a in one single-steel fiber. The PVA fibers are the more dispersed “peanut”-shaped sections in Figure 10b. In addition, a high density of Ca and Si elements around the fibers shows well-hydrated areas of C-S-H, indicating a strong bond between the fibers and the cement mortar. However, it is worth noting that in the steel fiber image, the quartz sands seem to agglomerate around the fiber due to its relatively large size. In the PVA fiber images, quartz sands are more evenly distributed among the fibers, and the EDS mapping indicated a more homogenous commingle of sands and C-S-H. This finding could support the specific strength results where the mixture with a higher PVA fiber dosage demonstrated similar specific strength with the mixture with a low steel fiber dosage. 

SE-SEM images of the post-fracture (flexural test) surface of the specimen are shown in Figure 11. In Figure 11a, it can be found that the steel fibers were bent after the test, indicating a strong bond between the steel fibers and the cement mortar, as well as a higher energy absorption at failure. Parallel scratches shown in Figure 11b on the surface of the steel fiber, which were caused likely due to fiber pull out, are another evidence of a strong bond. In addition, the residual hydration products on the fiber also indicate a strong adhesion between the steel fibers and the cement mortar. All steel fibers observed in the SEM images remained intact without obvious fracture, indicating the energy absorption is mainly due to the plastic deformation of the steel fiber and the friction during fiber pull out. In Figure 11c, cracks in the mortar around the steel fiber indicate anchorage failures during fiber pull out, which is also proof that the presence of the still fiber inhibited the cracks through the bridging effect [32]. This finding is consistent with the conclusion of the BSE-SEM image of the existence of a weak region around the steel fiber due to quartz sand agglomeration.

In Figure 11d, it can be seen that the PVA fiber is fractured, as indicated by an uneven end section and necking. The PVA fiber underwent a large amount of deformation and eventually exceeded its strain limit under overstressed conditions, which is likely the reason why the compressive strength of the PVA fiber-reinforced concrete is lower than that of steel fiber-reinforced concrete. By observing Figure 11e, it can be found that there is a large amount of hydration products remaining on the PVA fiber surface, indicating a strong bond during fiber pull out. Unlike steel fiber-reinforced concrete, PVA fibers demonstrated fewer anchorage failures (Figure 11d–f) but more fiber fractures, which likely will enable PVA fibers to have better control of the crack expansion. 

In sum, by comparing the fracture surfaces of two kinds of fibers, it can be concluded that the roles of steel fiber and PVA fiber are different in the reinforced cement matrix. For steel fiber, the energy absorption is mainly from the plastic deformation of the steel fiber and friction during fiber pull out due to a much higher strength/modulus of elasticity with a larger diameter of the steel fiber. However, PVA fibers are effective due to the crack-bridging effect through their own ductility and plastic deformation to absorb energy to delay cracks. The next section will introduce a more comprehensive method to evaluate the fiber performance in order to provide guidance on how to effectively select fiber reinforcement for UHPC.

### 3.5. Performance Evaluation

The radar charts and radar area diagrams of all UHPCs are shown in Figure 12. The performance is evaluated in five criteria: compressive strength, flexural strength, absorbed energy, specific strength, and fiber dosage. By calculating the area of the constituent pentagons, the performance of all UHPCs can be quantitively evaluated. Among them, fiber dosage was included as one criterion in order to make lower dosage more favorable. As shown in Figure 12c, most of the PVA fiber-reinforced concretes obtained improved performance, with only a decrease in performance when a 6 mm length was selected. The performance of steel fiber-reinforced concrete is much higher, indicating that in most cases, the performance improvement of the concrete resulting from the selection of steel fibers is superior to that of PVA fibers. However, it is worth noting that the performance enhancement of concrete with steel fibers and Type II 12 mm PVA fibers at 0.5% is very close to each other. Further observation in the radargram in Figure 12b reveals that SF 0.5 is only slightly better than Type II 12 mm 0.5 in terms of absorbed energy and flexural strength but lower in compressive and specific strengths, which shows that PVA fiber has the potential to replace steel fiber at a lower dosage.

Comparing the performance of the two kinds of fibers at low dosages, the steel fiber is superior in improving toughness, while the advantage of PVA fibers results in higher specific strength. For specimens that require high stress levels, steel fiber is recommended. 

For underdressed conditions, PVA fibers can be an alternative reinforcement to steel fiber in UHPC. 

## 4. Conclusions

In this study, the performance of different dosages, lengths, and surface treatments of PVA fibers was investigated through a comprehensive evaluation. A few conclusions can be drawn as follows. 

(1) At the dosage of 0.5 vol% of, with the selection of 12 mm Type II PVA fiber, the mechanical performance of resulting UHPC reached 158.97 MPa in compressive strength and 24.86 MPa in flexural strength, respectively with an increase of 12.0% and 6.0% to the control UHPC mixture without fiber.

(2) The incorporation of all types of PVA fibers increased the energy of destruction (up to 18.6%), and most of them led to increased specific strength (up to 12.7%).

(3) Microstructure analysis revealed that the enhancing mechanisms of steel fiber and PVA fiber are different. Steel fiber works better in improving toughness due to its high strength and modulus of elasticity, while PVA fibers are effective due to the crack-bridging effect through their own ductility and plastic deformation to absorb energy to delay cracks.

(4) Based on a comprehensive evaluation, modified PVA fiber (12 mm Type II 0.5%) has the potential to replace steel fiber at a lower dosage (0.5%).

## Figures and Tables

**Figure 1 polymers-16-03449-f001:**
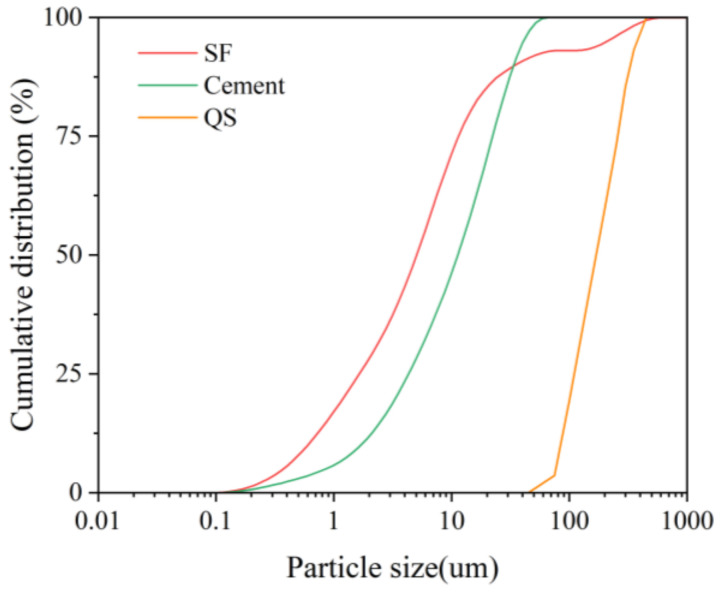
Particle size distributions of cement, silica fume (SF), and quartz sand (QS).

**Figure 2 polymers-16-03449-f002:**
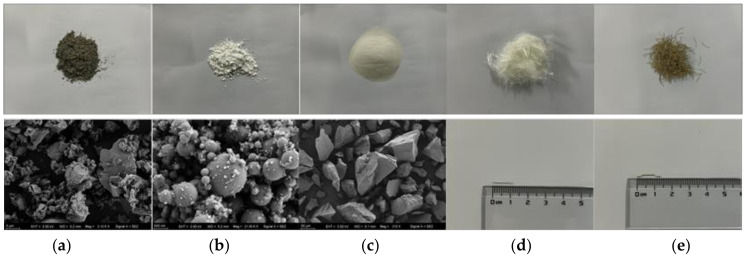
Images of materials used: (**a**) cement (scale bar is 5 μm), (**b**) silica fume (scale bar is 500 nm), (**c**) quartz sand (scale bar is 50 μm), (**d**) PVA fibers, and (**e**) steel fibers.

**Figure 3 polymers-16-03449-f003:**
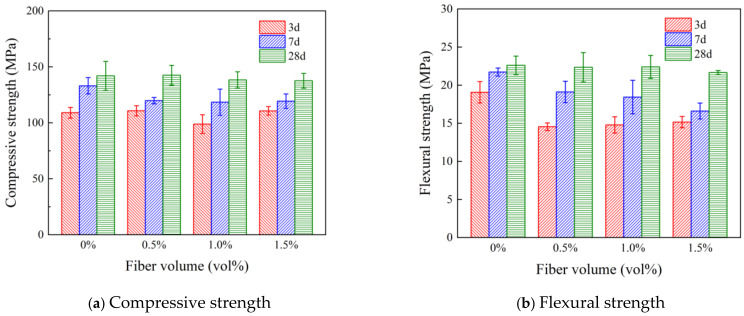
Effect of PVA fiber dosage on the mechanical properties of UHPC.

**Figure 4 polymers-16-03449-f004:**
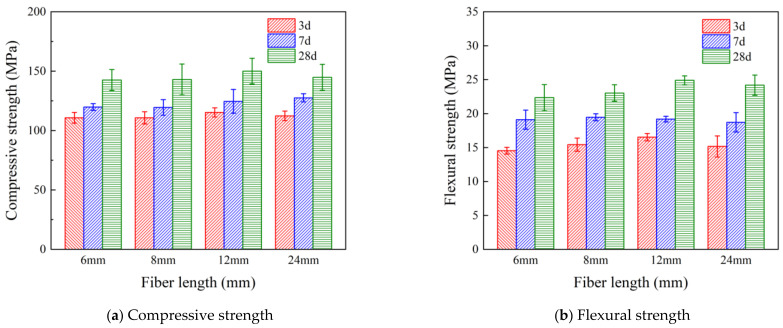
Effect of PVA fiber length on the mechanical properties of UHPC.

**Figure 5 polymers-16-03449-f005:**
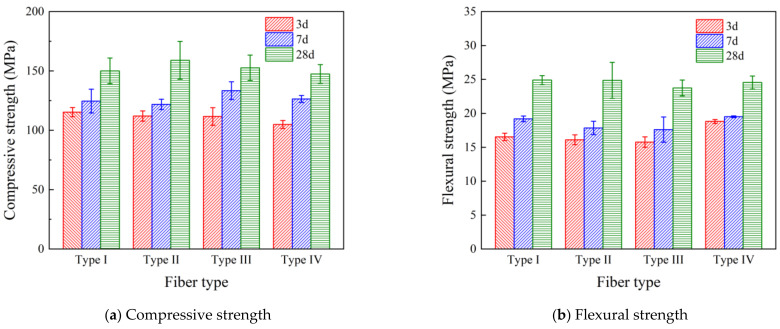
Effect of PVA fiber treatment on the mechanical properties of UHPC.

**Figure 6 polymers-16-03449-f006:**
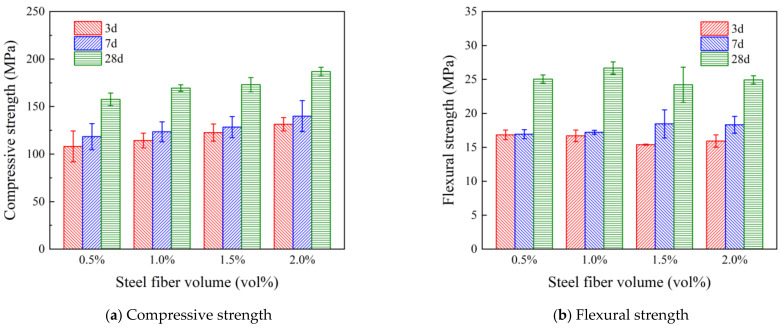
Effect of steel fiber dosage on the mechanical properties of UHPC.

**Figure 7 polymers-16-03449-f007:**
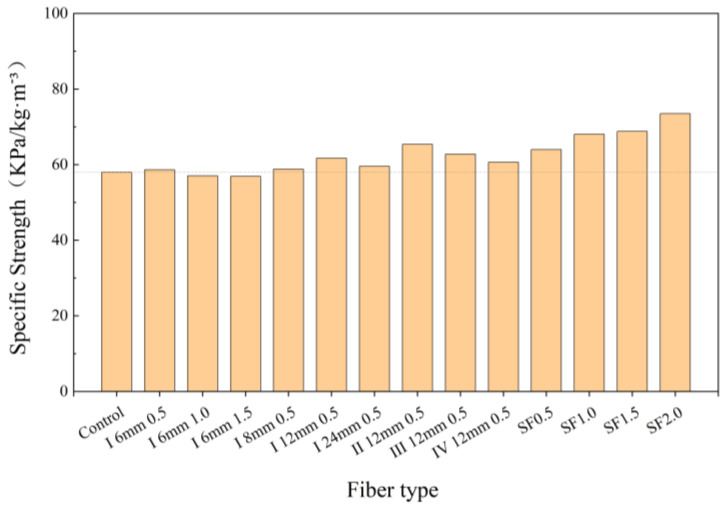
Specific strengths of UHPC.

**Figure 8 polymers-16-03449-f008:**
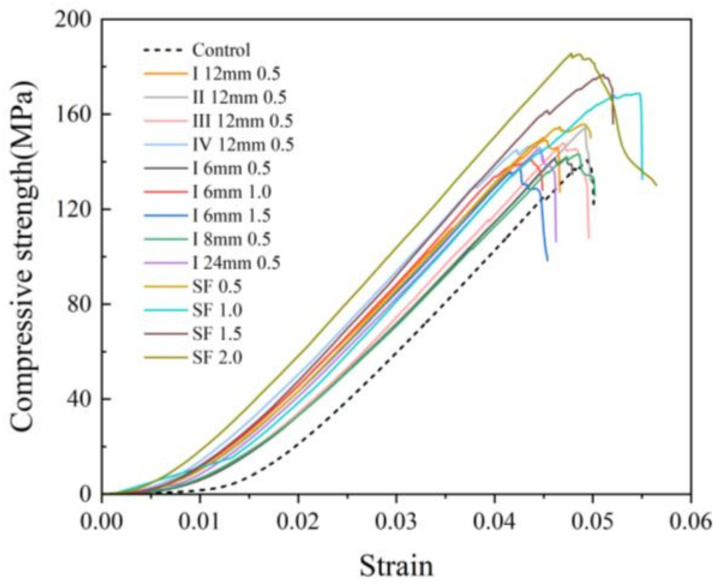
Stress–strain curve of compressive strength.

**Figure 9 polymers-16-03449-f009:**
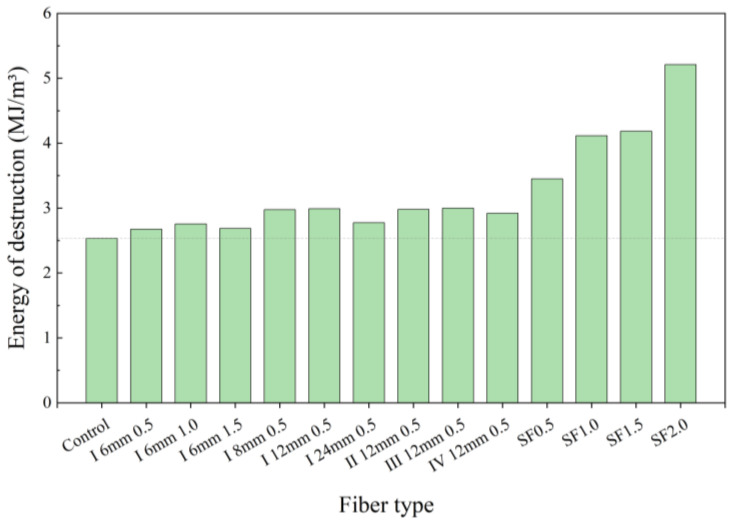
Energy absorption capacity.

**Figure 10 polymers-16-03449-f010:**
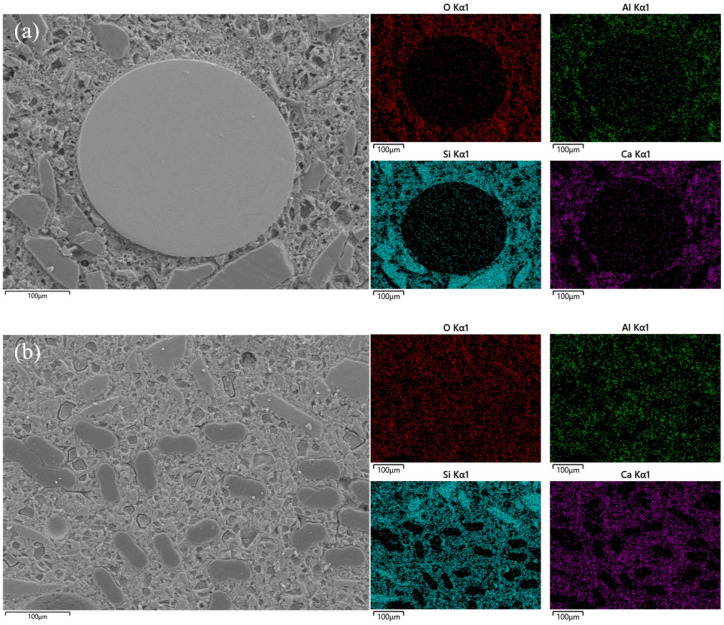
BSE-SEM images and corresponding EDS mapping of different samples after 28 d of curing: (**a**) SF 2.0; (**b**) PVA Type II 12 mm 0.5.

**Figure 11 polymers-16-03449-f011:**
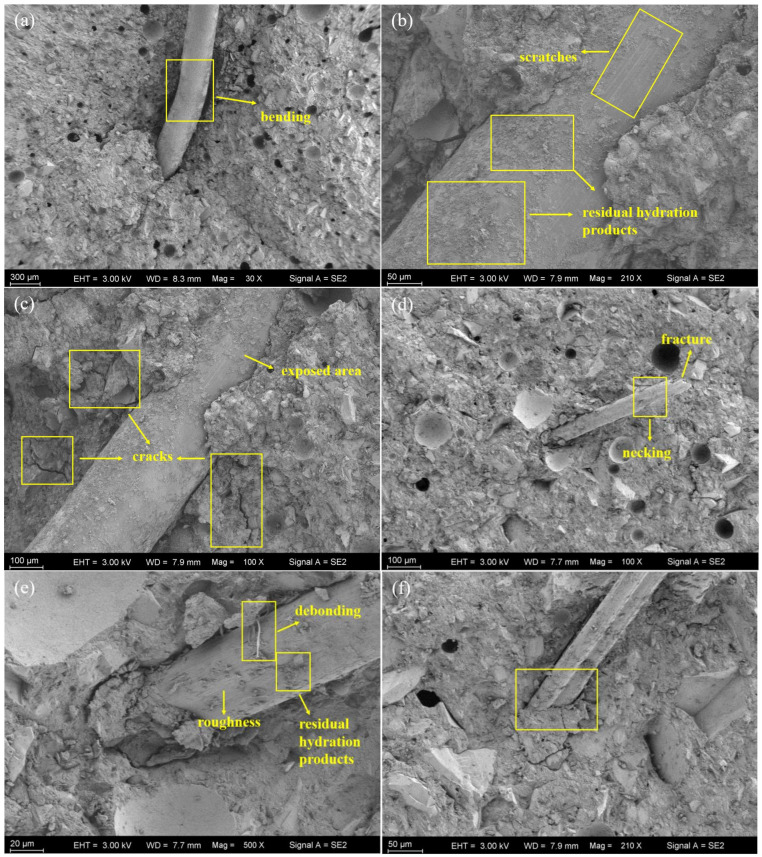
SEM images of the fracture surface of (**a**–**c**) SF2.0 (**d**–**f**) II 12 mm 0.5.

**Figure 12 polymers-16-03449-f012:**
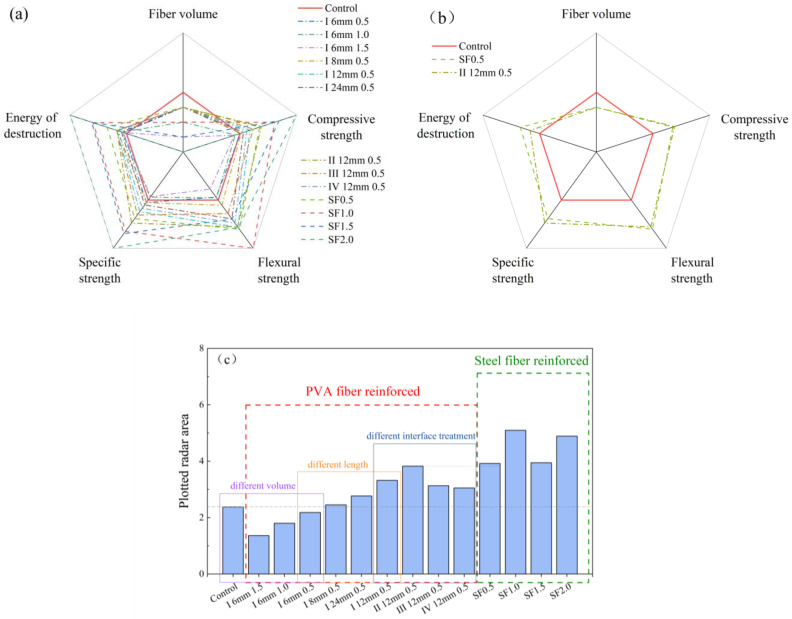
(**a**,**b**) Radar chart diagram; (**c**) plotted radar area of (**a**).

**Table 1 polymers-16-03449-t001:** The chemical composition of materials (%).

	CaO	SiO_2_	Al_2_O_3_	SO_3_	Fe_2_O_3_	MgO	K_2_O	TiO_2_
Cement	64.3	18.6	5.84	3.62	2.99	2.90	0.69	0.35
SF	0.06	91.6	0.09	0.34	0.28	0.06	0.05	0.04
QS	0.07	99.8	0	0	0.02	0.07	0.007	0.002

**Table 2 polymers-16-03449-t002:** Physical properties and dimensions of the fibers.

	Length (mm)	Diameter (μm)	Tensile Strength (MPa)	Modulus of Elasticity (GPa)
Type I	6, 8, 12, 24	40	1500	38
Type II	12	40	1500	38
Type III	12	40	1500	38
Type IV	12	40	1500	38
Steel fiber	13	220	2850	200

**Table 3 polymers-16-03449-t003:** Mixture proportions of control UHPC (kg/m^3^).

	Cement	Silica Fume	Quartz Sand	Water	Superplasticizer
Control	1363	342	402	180	44

**Table 4 polymers-16-03449-t004:** Fiber types and dosages (kg/m^3^).

	6 mm Type I PVA	8 mm Type I PVA	12 mm Type I PVA	24 mm Type I PVA	12 mm Type II PVA	12 mm Type III PVA	12 mm Type IV PVA	Steel Fiber
I 6 mm 0.5	6.2	\	\	\	\	\	\	\
I 6 mm 1.0	12.4	\	\	\	\	\	\	\
I 6 mm 1.5	18.6	\	\	\	\	\	\	\
I 8 mm 0.5	\	6.2	\	\	\	\	\	\
I 12 mm 0.5	\	\	6.2	\	\	\	\	\
I 24 mm 0.5	\	\	\	6.2	\	\	\	\
II 12 mm 0.5	\	\	\	\	6.2	\	\	\
III 12 mm 0.5	\	\	\	\	\	6.2	\	\
IV 12 mm 0.5	\	\	\	\	\	\	6.2	\
SF 0.5	\	\	\	\	\	\	\	37.4
SF 1.0	\	\	\	\	\	\	\	74.9
SF 1.5	\	\	\	\	\	\	\	112.3
SF 2.0	\	\	\	\	\	\	\	149.8

## Data Availability

The raw data supporting the conclusions of this article will be made available by the authors on request.

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
