# Peer review of "Influence of Modified PVA Fiber on Ultra-High Performance Concrete and Its Enhancing Mechanism"

_polymers, 2024, doi:10.3390/polym16233449_

Round 1

Reviewer 1 Report

Comments and Suggestions for Authors

Observations

1. First of all, the introduction part must be completed significantly in the direction of composition-properties-applications of UHPC. Also, it should be very clearly pointed out what are the advantages of using PVA fibers compared to metallic ones, from the point of view of the field of use;

2. In fig. 2, the images must be noted, and for the electron microscopy pictures, the magnification must be specified;

3. From an experimental point of view, subchapter 2.1 Materials and Mixture Proportions must be completed with information on the workability of mixtures; the influence of fiber addition on workability and the water-cement ratio; explaining the use of such a high percentage of fumed silica (25% by mass compared to cement), which is unusual; explanation of dosages 0.5%; 1%; 1.5% compared to what?

4. The bibliography must be completed.

Author Response

The authors would like to thank the reviewer for the comments to improve the manuscripts. Changes have been made accordingly. Responses are detailed as follows.

 Comment 1: First of all, the introduction part must be completed significantly in the direction of composition-properties-applications of UHPC. Also, it should be very clearly pointed out what are the advantages of using PVA fibers compared to metallic ones, from the point of view of the field of use.

Response 1: As suggested, the introduction is now significant improved with a dozen more recent literatures referenced. Additional introduction is also added explaining the objectives of this study highlighting the advantages of using PVA fibers instead of steel fiber in certain field applications.  

Comment 2: In fig. 2, the images must be noted, and for the electron microscopy pictures, the magnification must be specified;

Response 2: Revised as suggested.

Comment 3: From an experimental point of view, subchapter 2.1 Materials and Mixture Proportions must be completed with information on the workability of mixtures; the influence of fiber addition on workability and the water-cement ratio; explaining the use of such a high percentage of fumed silica (25% by mass compared to cement), which is unusual; explanation of dosages 0.5%; 1%; 1.5% compared to what?

Response 3:

  • Workability was tested using the “mini-slump test” which was done using a “mini” bronze slump cone (60 mm in height with a 36 mm top opening diameter and 60 mm bottom diameter). From preliminary tests, we found that a 150 mm spread size was a decent value for the UHPC to have a satisfactory workability (flow from mixer to mold without difficulty). Therefore, superplasticizers were added and adjusted as necessary to reach at least 150 mm spread size for all mixtures. Changes were made in the manuscript to clarify the workability issue.
  • Mixture design for Control UHPC used in this study was based on Huang’s doctoral dissertation (ref [30]). A 25% silica fume was unusually high for normal concrete, but quite common for UHPC.
  • All dosages were based on volume. This was made clear in the text.

Comment 4: The bibliography must be completed.

Response 4: Revised as suggested.

Reviewer 2 Report

Comments and Suggestions for Authors

The manuscript can be accepted after considering the following comments:

1. The most important results should be added in the abstract.

2. The introduction is very limited and need to be rewritten.

3. Grain size distribution of the used material should be discussed.

4. The objectives of the experimental program should be discussed in details.

5. Test setup and measurements should be added.

6. What is the effect of fiber type on splitting tensile strength?

7. What is the effect of fiber type on stiffness of concrete mixtures

8. Current results showed be compared with previous findings

9. Conclusions is poorly written.

10. Recent references should be added.

Author Response

The authors would like to thank the reviewer for the comments to improve the manuscripts. Changes have been made accordingly. Responses are detailed as follows.

The manuscript can be accepted after considering the following comments:

  1. The most important results should be added in the abstract.

Response: Changed as advised.

  1. The introduction is very limited and need to be rewritten.

Response: As suggested, the introduction is improved in the direction of composition-properties-applications of UHPC with fiber reinforcement. In addition, the advantages of using PVA fibers instead of steel fibers are pointed out, from the perspective of field applications.

  1. Grain size distribution of the used material should be discussed.

Response: Particle size distributions are shown in Figure 1. The nominal maximum grain size for quartz sand is 0.6 mm. The caption of Figure 2 was improved with scale bar indications. And more discussion was added following Figure 2.

  1. The objectives of the experimental program should be discussed in details.
  2. Test setup and measurements should be added.

Response: New subchapters (2.2.1 to 2.2.4) were added to describe the objectives and setup for the experiments.

  1. What is the effect of fiber type on splitting tensile strength?

Response: Splitting tensile strength was not tested in this study. Instead, flexural strength was tested, which was often used as an alternative test to evaluate the performance under tension. The results are shown in Figure 5 and discussed in Chapter 3.1.3 PVA fiber treatment.

  1. What is the effect of fiber type on stiffness of concrete mixtures

Response: If understood correctly, “stiffness” herein means modulus of elasticity. According to the manufacturers, the treatment of the PVA fiber is restricted to the “surface modification”, hence the fiber type would impact the modulus of elasticity quite insignificantly.

  1. Current results showed be compared with previous findings

Response: Several results now are compared with references (see ref [27, 31, 32]).

  1. Conclusions is poorly written.

Response: Rewritten as suggested.

  1. Recent references should be added.

Response: Changed as suggested. A dozen new and more recent references were added where necessary. Please refer to the updated reference list.   

Round 2

Reviewer 1 Report

Comments and Suggestions for Authors

I belive that the work can be published in this form.

Reviewer 2 Report

Comments and Suggestions for Authors

The paper can be accepted